# Role of ferroelectric polarization during growth of highly strained ferroelectric materials

Rui Liu [1], Jeffrey G. Ulbrandt[2], Hsiang-Chun Hsing[1], Anna Gura[1], Benjamin Bein[1], Alec Sun[1], Charles Pan[1], Giulia Bertino[1], Amanda Lai[1], Kaize Cheng[1], Eli Doyle[1], Kenneth Evans-Lutterodt[3], Randall L. Headrick [2] & Matthew Dawber[1✉]

In ferroelectric thin films and superlattices, the polarization is intricately linked to crystal structure. Here we show that it can also play an important role in the growth process, influencing growth rates, relaxation mechanisms, electrical properties and domain structures. This is studied by focusing on the properties of $BaTiO_3$ thin films grown on very thin layers of $PbTiO_3$ using x-ray diffraction, piezoforce microscopy, electrical characterization and rapid in-situ x-ray diffraction reciprocal space maps during the growth using synchrotron radiation. Using a simple model we show that the changes in growth are driven by the energy cost for the top material to sustain the polarization imposed upon it by the underlying layer, and these effects may be expected to occur in other multilayer systems where polarization is present during growth. This motivates the concept of polarization engineering as a complementary approach to strain engineering.

[1] Department of Physics and Astronomy, Stony Brook University, Stony Brook, NY 11794-3800, USA. [2] Department of Physics and Materials Science Program, University of Vermont, Burlington, VT 05405, USA. [3] Brookhaven National Laboratory, Upton, NY 11973, USA. ✉email: matthew.dawber@stonybrook.edu

Strain engineering of perovskite oxide thin films has proven to be an extremely powerful approach for enhancing and inducing ferroelectric behavior. Some of the most notable achievements in recent years have included the enhancement of polarization in strained BaTiO₃ by 270%[1], discovery of a super-tetragonal phase of BiFeO₃[2] and a strain-enabled electric control over magnetization in EuTiO₃[3,4]. In ferroelectric thin films and multilayers, the polarization is intricately linked to crystal structure, so that strain and electrostatic boundary conditions have considerable impact on the magnitude of the polarization and the arrangement of polarization domains[5–21]. In certain strained ferroelectrics, for example, BaTiO₃ (BTO) or PbTiO₃ (PTO) grown epitaxially on SrTiO₃ (STO), the ferroelectric transition temperature can lie above the growth temperature of the film[16,22].

We demonstrate that when this is the case it is possible to engineer material properties of ferroelectric thin films during growth, not only by strain, but also through polarization. We provide insight into the mechanism through which this occurs by performing in situ x-ray diffraction during growth at the National Synchrotron Light Source II (NSLS-II). By combining results from X-ray powder diffraction (XRD), piezoforce microscopy (PFM), electrical measurements, and especially in situ XRD at NSLS-II, the ferroelectric polarization underneath BTO films during growth is shown to help them stay in highly strained states. Polarization during growth is seen to play an important role in the growth process, influencing growth rates, relaxation mechanisms, electrical properties, and domain structures. This suggests polarization engineering is a powerful approach to the design of improved ferroelectric thin films.

## Results

**Growth rates.** In general growth rates for thin films are mostly determined by the incident flux of material and are thus not expected to vary greatly with temperature and over many years of growing superlattices containing PTO[11,12,23,24], we have not observed significant temperature dependence of the growth rate of one of the layers. In particular, we do not see significant temperature dependence of the growth rate when dielectric layers, such as SrTiO₃ or metallic layers such as SrRuO₃ are used. Further, when the growth rate of BTO thin films (≈20 nm thickness) at different temperatures (black data, Fig. 1a) is measured, it does not appear to depend on temperature. However, a marked demonstration of the effect of ferroelectric polarization during growth can be seen in BaTiO₃/PbTiO₃ (BTO/PTO) superlattices on SrTiO₃ substrates. Owing to the high compressive strain, particularly that imposed upon BTO, both BTO and PTO will have highly elevated ferroelectric transition temperatures, though this effect does compete with supression of their ferroelectriciy owing to depolarization field when they are ultrathin. Thus, for practical deposition temperatures, both of these materials will start out paraelectric and become ferroelectric at a thickness defined by the growth temperature as they are grown. Here we focus on a series of three unit cell (u.c.)/three unit cell BTO/PTO superlattices that were grown in the temperature range from 450 °C to 500 °C. The growth rates of the BTO layers were found to change dramatically over this temperature range (purple data, Fig. 1a). We thus suppose that the existence of the three u.c. layers of PTO has a key role in determining the BTO growth rate within the superlattice. Previous studies of ultrathin PTO films found that the ferroelectric phase transition temperature of three u.c. films of PTO is close to our growth temperature[10], so a plausible hypothesis is that the change in growth rate of BTO may be associated with the large changes of polarization with temperature that PTO should display in the vicinity of the ferroelectric phase transition.

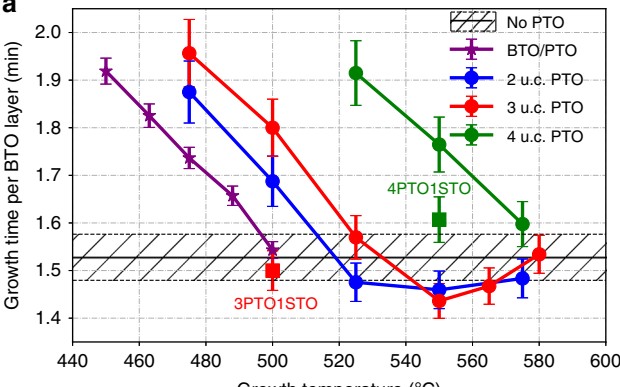

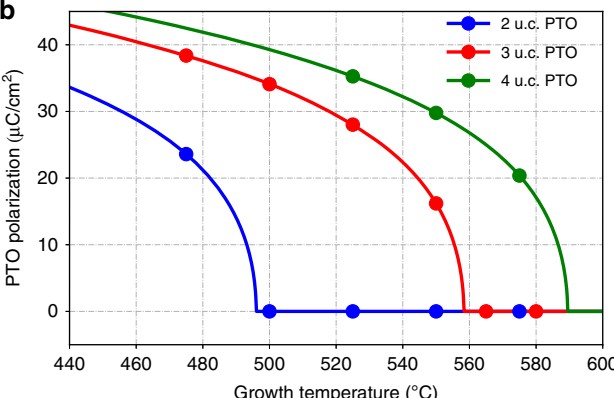

**Fig. 1 Growth rates of BTO and polarization of PTO changes vs temperature.** Growth rates and polarization associated with phase transition: **a** Growth time per BTO layer plotted against growth temperatures in 20-nm pure BTO films (black), 50 nm BTO/PTO superlattices (purple), 20 nm BTO films grown on ultrathin PTO films (circle data point markers, blue: 2 u.c. layers PTO, red: 3 u.c. layers PTO, green: 4 u.c. layers PTO), and BTO films with an additional 1 u.c. layer STO on top of PTO ultrathin films (square data point markers, red: 3 u.c. layers PTO, green: 4 u.c. layers PTO). Error bars reflect uncertainty in the thickness of the films as obtained by x-ray diffraction measurements and fitting **b** Polarization of ultrathin PTO films under growth conditions calculated according to Eq. (1). Source data are provided as a Source Data file.

To verify the hypothesis, ultrathin PTO films were grown at temperatures in the vicinity of the ferroelectric phase transition temperature and then BTO films were grown on top of them to a thickness of 20 nm. In other words, the BTO films were grown on substrates with various ferroelectric polarization values. A 20 nm thick SrRuO₃ (SRO) electrode was grown beneath all of the films to enable electrical measurements. The surface quality of the grown films was checked using atomic force microscopy (AFM) (Supplementary Fig. 2). Crystal structures and growth rates were determined by X-ray diffraction θ–2θ scans and low-angle reflectivity scans on a Bruker D8-Discover high-resolution X-ray diffractometer (Supplementary Fig. 1). Three series of samples were prepared with two, three, and four u.c. layer thick PTO films, and the measured average growth rates for these films as function of temperature in between 450 °C and 580 °C are shown in (Fig. 1a blue, red, and green). We did not examine growth above 600° as there is substantial Pb loss from the PTO at these higher growth temperatures. The growth rate displayed is the average growth rate for the growth of the entire 20 nm film and it can be seen that overall the growth rates are somewhat slower for

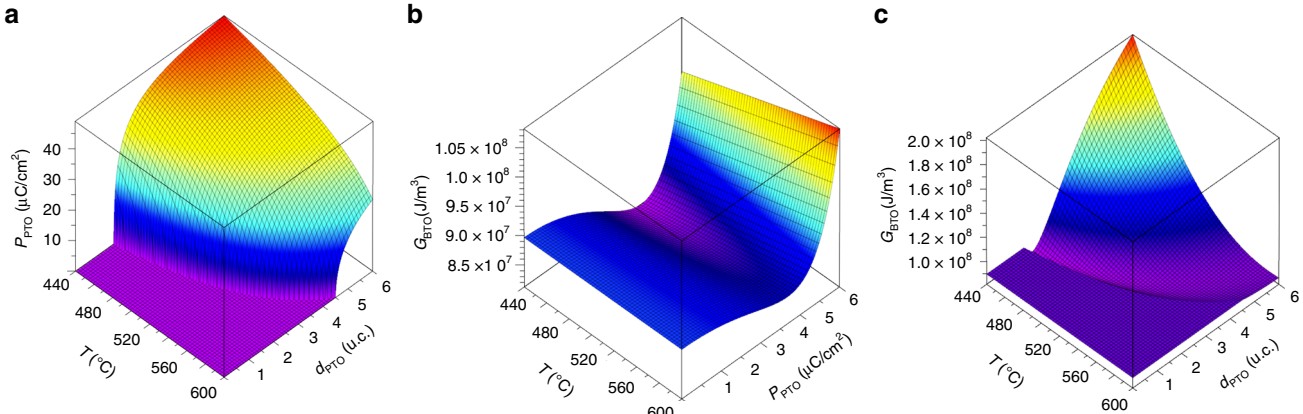

**Fig. 2 Explanation of growth rate through free energy of BTO.** Explanation of growth rate through free energy of BTO: **a** Polarization of PTO as a function of temperature and PTO thickness d, **b** Free energy of BTO as a function of temperature and polarization, which in our model is assumed to be the polarization of the underlying PTO, **c** Free energy of BTO as a function of temperature and PTO thickness d.

the films than the three u.c. grown in the BTO/PTO superlattices. This change in growth rate as the film thickness increases is confirmed by our in situ x-ray diffraction measurements (Supplementary Fig. 6)

To understand the link between the growth temperature and PTO thickness and the ferroelectric polarization of PTO at the growth condition, the Landau approach of Pertsev et al.[25] was used. In this approach, the out-of-plane polarization of the ultrathin strained PTO can be calculated from minimizing

$$ G = a_3^* P^2 + a_{33}^* P^4 + a_{111} P^6 + \frac{u_m^2}{s_{11}+s_{12}} + \frac{\lambda_{eff}}{d\epsilon_0} P^2 \qquad (1) $$

where $a_3^* = a_1 - u_m \frac{2Q_{12}}{s_{11}+s_{12}}$ and $a_{33}^* = a_{11} + \frac{Q_{12}^2}{s_{11}+s_{12}}$.

We have used the effective screening length $\lambda_{eff}$ as an adjustable parameter to approximately match the transition temperature observed in experiment; the value we used for all calculations shown in the paper was $5 \times 10^{-13}$ m. The predicted polarization of the PTO film under growth conditions is shown in Figs. 1b and 2a.

The temperature dependence of growth time per BTO layer in each series (Fig. 1a) shows a strong correlation with the expected polarization of PTO during the growth (Fig. 1b), suggesting that the growth rate of the BTO film changes with the ferroelectric polarization of PTO film on which it grows. If one considers the point at which growth rates begin to depend on temperature as the ferroelectric transition temperature we see that the transition temperature increases as the film becomes thicker, which has been confirmed previously[8]. The growth rate of the BTO film does not change when the PTO film under it is in the paraelectric state during growth, whereas it takes longer to grow when the ferroelectric polarization of PTO film increases. A simple way for rationalizing this result is to consider the additional energy required to change the thickness of the film. Assuming all other things are equal, the additional polarization free energy to add a unit cell of BTO is directly proportional to $G_{BTO}$ (Fig. 2b). The initial boundary condition we impose on BTO growing on PTO is that the polarization is continuous across the interface. We therefore take the calculated polarization for the thin PTO layer $P_{PTO}$ (Fig. 2a) as the polarization of BTO and calculate $G_{BTO}$ as a function of temperature and the thickness of PTO, d (Fig. 2c) using again the approach of Pertsev et al.[25] Although the approximation of continuous polarization will quickly break down as the layer grows requiring additional terms to be added to the expansion[26], we use the continuous polarization approach to

find the simplest way to understand the observed slow down of the growth rate.

Two considerations are important in elevating the bulk ferroelectric free energy of BTO to be a significant factor in the growth, compared with the normally more relevant surface and interface energies, one is the sharp increase in $G_{BTO}$ as a function of $P$ once it exceeds the optimal value for BTO, and the other is the baseline increase in free energy to $\sim 1 \times 10^8$ J/m³ owing to the mismatch strain imposed on BTO.

Our calculation reveals that there is actually a small region of parameter space close to the phase transition of PTO where polarization is fairly small and there is a decrease in energy that we might expect leads to a faster growth rate. However, for larger polarizations at lower temperatures and larger PTO thicknesses, the free energy of BTO is considerably higher as the PTO tries to impose a higher polarization on the BTO than ideal, resulting in an additional energy cost to increase the thickness of the film. Careful inspection of Fig. 1a reveals that it is precisely the behavior which is experimentally observed.

In principle, the slow down in growth rate for large PTO polarization is not reliant on the ferroelectric nature of BTO, a dielectric material should similarly experience an energy penalty when polarization is imposed on it. However, for the frequently studied case of SrTiO₃ grown on PTO on SrTiO₃ substrates, the energy penalty for the polarization values considered here is an order of magnitude less, owing to SrTiO₃ having a large dielectric constant and the lack of a strain-induced increase in the bulk free energy, and thus there is not a significant change in the growth rate of SrTiO₃ owing to imposed polarization, which is what we have observed in experiment. However, for dielectric layers with lower dielectric constant, or under significant strain we should expect this effect to be observed.

Further evidence for our hypothesis was obtained by growing films in which a single unit cell of STO was grown in between the PTO and the BTO, and these films show a marked reduction in the growth time per layer, as expected by the reduction of polarization that should occur owing to the insertion of the STO layer. The connection between BTO growth rate and the ferroelectric polarization of the PTO film underneath also provides an explanation for why the growth rate of BTO film changes with temperature in BTO/PTO superlattices, whereas it remains stable in pure BTO films.

**Ferroelectric properties.** The post-deposition functional properties of the films were measured at room temperature. One very intriguing observation is that the room temperature out-of-plane

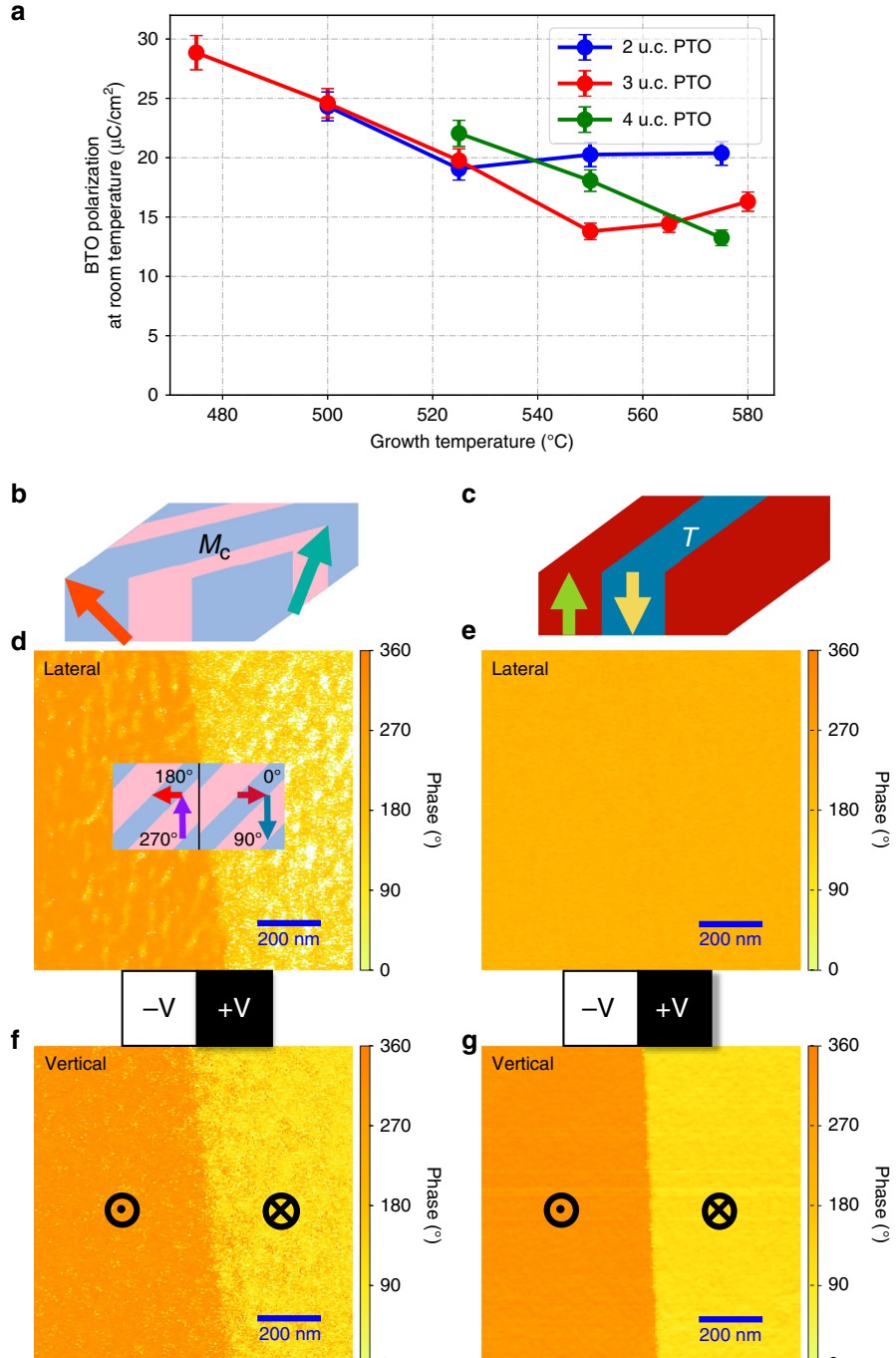

**Fig. 3 Lateral and vertical PFM on films grown on ferroelectric/paraelectric PTO and schematic views of the ferroelectric domain structures.** Post-deposition polarization properties of films: **a** Polarization of BTO films grown on ultrathin PTO films measured at room temperature (blue: 2 u.c. layers PTO, red: 3 u.c. layers PTO, green: 4 u.c. layers PTO). **b**–**g** Ferroelectric domains measured by vector PFM and their possible arrangements: schematics of Monoclinic $M_c$ (**b**) and Tetragonal (**c**) domain structures of BTO films grown on paraelectric/ferroelectric three u.c. layers PTO films, respectively. Two stable states of uniform and opposite polarization, up and down, were achieved by applying negative voltage in the out-of-plane direction on the left half side and positive voltage in the out-of-plane direction on the right half side of each film. The larger micron scale domain structure is that written with the AFM tip. The smaller scale texture in the images is from the naturally occurring domains which are on the 20–60 nm length scale. In contrast with vertical response (**f**, **g**) there is no lateral phase PFM response (**e**) for the film grown on ferroelectric PTO. On the other hand, the lateral cross section of domains (inserted image in **d**) in the film grown on paraelectric PTO indicates two polarization orientations with 90° domain walls in each state, and the lateral phase PFM image (**d**) does show the 180°/270° (up state) and 0°/90° (down state) domain configuration with four orientations. Source data are provided as a Source Data file.

polarization of the film as measured by hysteresis measurements also depends on the magnitude of the PTO polarization during growth (Fig. 3a). Taking films with three u.c. layers of PTO as an example (red data, Fig. 3a), the out-of-plane polarization of a BTO film grown on ferroelectric PTO simultaneously increases with PTO polarization and can be as large as doubled (29 μC/cm²) compared with the one of BTO film grown on paraelectric PTO film (14 μC/cm²). Vector PFM was then performed in order to

determine the piezoelectric response in micro-scale in both lateral and vertical directions. The study of two BTO films grown on three u.c layers paraelectric/ferroelectric (grown at 550 °C/500 °C) PTO is presented in Fig. 3. Two stable states of uniform and opposite polarization, up and down, were achieved by applying negative vertical voltage on the left and positive vertical voltage on the right. In the vertical phase PFM phase response (Fig. 3f, g), the two films displayed the same behavior; they have a uniform polarization direction in each up or down state and the two states have 180° difference in domain orientations. However, different behaviors were seen in the lateral PFM phase response. No lateral piezoelectric response (Fig. 3e) was observed in the films grown on ferroelectric PTO, whereas complex domain patterns (Fig. 3d) were seen in the other film grown on paraelectric PTO. In each of the up or down state, two types of polarization domains with 90° domain walls were observed, and the orientations of polarization in each domain were flipped, whereas the state changed from up to down. The 3D domain configuration inferred by combining the lateral and vertical PFM response is that the BTO film grown on paraelectric PTO presents low symmetry monoclinic $M_c$ polarization domains, while tetragonal polarization domains were observed in the BTO film grown on ferroelectric PTO. An additional observation made during the PFM measurement is that those films grown on ferroelectric PTO substrates have an electromechanical resonance frequency ~10% lower than those grown on paraelectric substrates (Supplementary Fig. 5), implying a difference in the elastic properties of the samples, which in turn have an influence on the electromechanical resonance frequency.

Owing to the large compressive strain imposed on a BTO film (−2.69%) grown on STO substrate, it is no surprise that the strained BTO film starts to relax at some point during the growth, forming some in-plane polarization domains. However, our experimental results suggest the ferroelectric polarization of the PTO film underneath can prevent or delay this relaxation and help the BTO film stay in a strained state, protecting the tetragonal polarization domains.

**In situ x-ray diffraction during growth**. To verify our hypothesis and gain more insight into the mechanisms at play, we performed a series of in situ x-ray diffraction experiments during growth of the model PTO/BTO system at the 4-D beamline at the NSLS-II synchrotron at BNL. These kinds of experiments are a powerful tool for gaining insight in to thin film growth in general[27–33] and more specifically the evolution of polarization and domain structure in growing ferroelectric films[10,16,22,34]. Figure 4c shows the calculated polarization from Eq. (1) for the thin PbTiO₃ layer $P_{PTO}$ plotted as a function of thickness and temperature, and provides a map to understand the parameters we chose for the PTO layer in our in situ growth experiments. BTO films of 24 nm thickness were grown on ultrathin PTO films with different thickness and different growth temperatures. Both the PTO and BTO were deposited while x-ray diffraction was performed. All films were grown on STO substrates with a 20 nm SRO electrode grown ex situ. Prior to the main set of experiments discussed here the evolution of the anti-Bragg peak $(0\,0\,\frac{1}{2})$ (to minimize bulk Bragg diffraction)[27,34,35]) was measured to calibrate the growth rates for both BTO and PTO films (Supplementary Fig. 6). In the following experiments, the same scanning technique was used as that of Bein et al.[16] for the study of ferroelectric-dielectric BaTiO₃/SrTiO₃ superlattices. The technique allows us to achieve a rapid reciprocal space map in 15s, which is much shorter than the time taken to deposit a single unit cell of material (a few minutes). In each scan, the in-plane angle $\phi$ moves continuously through a given angular range, which corresponds to perform a rocking curve. Thus, the measured intensity of each pixel is the integrated intensity over the rocking curve at that pixel. Accordingly one in-plane momentum transfer direction ($Q_y$) is integrated, whereas diffraction information in the other two directions ($Q_x$ and $Q_z$) can be obtained. Further explanation of the technique can be found in Methods and Supplementary Information.

The idea that the ferroelectric polarization of PTO carries over to BTO is a central part of our model. Depending on the electrical boundary conditions underneath them, Fong et al.[9,10] found that ultrathin PTO layers may have a stripe domain structure, which produces diffraction features. Specifically, they found that films grown on STO had a stripe domain structure, whereas, in their case, films grown with a SRO bottom electrode were monodomain. In our experiment when we performed (0 0 1) maps during the growth of BTO on PTO layers with different ferroelectric polarization states, we found that the scattering from stripe domains is strongly dependent on the thickness of the PTO layer and the growth temperature. We infer that, in contrast to the findings of Fong et al., our PTO films develop stripe domain structure even when they are grown on SRO electrodes. This difference is most likely due to the quite different growth processes used in our experiments (sputtering) and those of Fong et al., resulting in different electrostatic boundary conditions for the two cases. Reciprocal space maps made when 10 nm BTO films were grown on different PTO substrates are presented in Fig. 4a. The calculated polarization under growth conditions of PTO substrates for each film in Fig. 4a corresponds to the red circles in the plot of PTO thickness vs Growth temperature BTO films grown on ferroelectric substrates show clear and strong domain scattering, whereas no obvious domain scattering was observed when the films are grown on paraelectric substrates. To ensure that the scattering observed was in fact related to ferroelectric domain structure, some of these films had top electrodes deposited on them post-deposition and we were able to verify that the domain scattering responded to an applied electric field (see Supplementary Fig. 11.) Two pure BTO films without PTO layers (labeled as 0 uc were grown at different temperatures. The similarity of the domain scattering from these samples with the ones grown on paraelectric substrates suggest again that this phenomenon is caused by ferroelectric polarization rather than purely temperature. The domain size evolution during the growth is plotted in Fig. 4b. The procedure by which the domain size is calculated is outlined in section H. of the Supplementary Information (Supplementary Fig. 8) . The domain sizes of BTO films grown on ferroelectric substrates stabilized faster as the ferroelectric domains of PTO are carried over to BTO, whereas the ones grown on paraelectric substrates go through a more dramatic evolution.

To study strain relaxation we carried out another set of maps, this time around the (1 0 1) Bragg peak. Figure 5a shows close-ups of the relaxed feature on a number of samples. These figures have been arranged according to their position on a plot of thickness vs growth temperature as shown in Fig. 5b, c. As in Fig. 4, the polarization of PTO substrates of each films in Fig. 5a corresponds to the red circles in Fig. 5b. The thickness oscillations in these images correspond to the strained part of the BTO film, which is constrained in plane to the STO substrate. In addition to this, we observe a diffraction feature associated with relaxed BTO. The relaxed parts of the BTO films grown on ferroelectric PTO is connected with the strained BTO with a continuous tail, whereas the one grown on paraelectric PTO is separated from the strained part. The

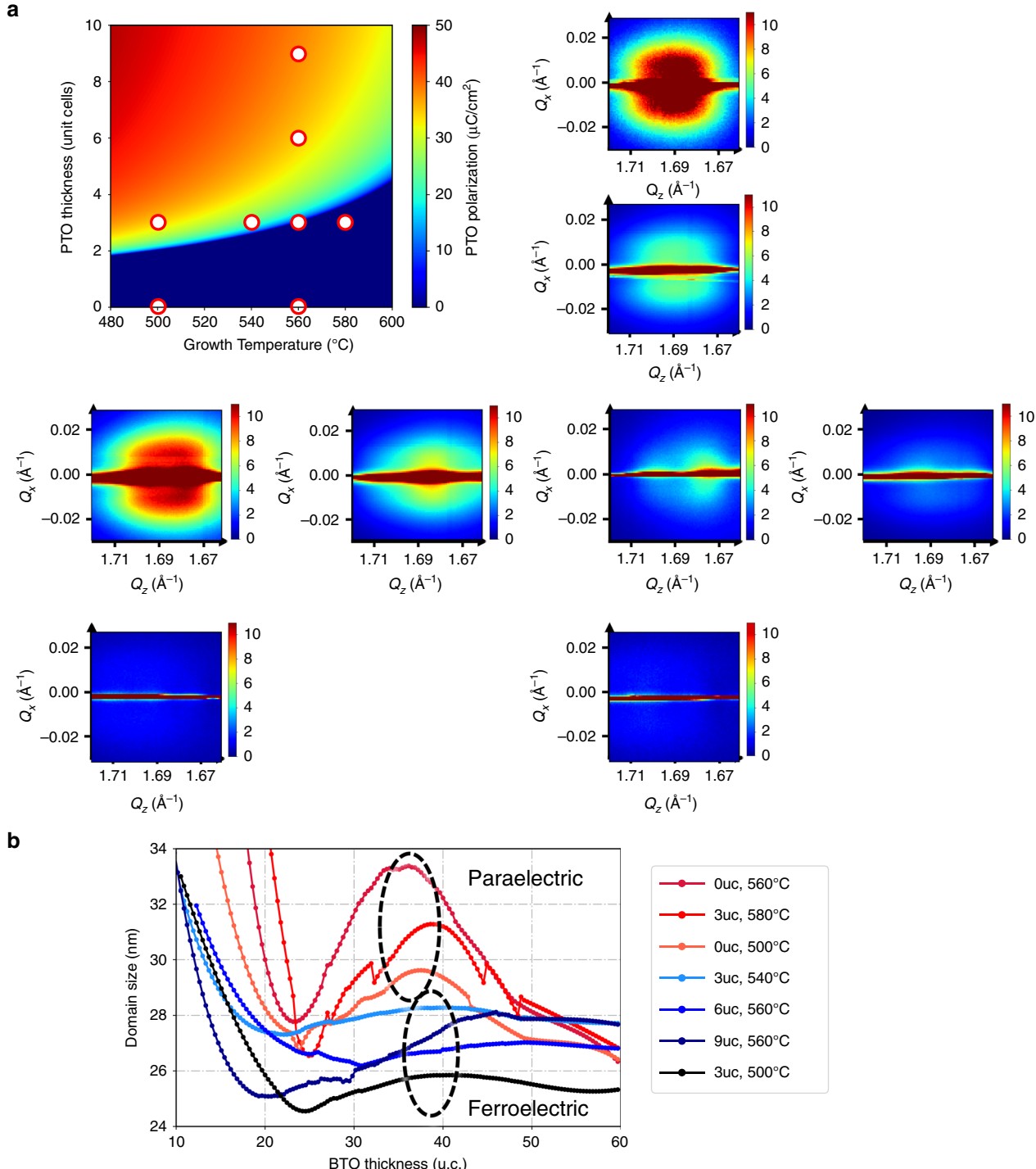

**Fig. 4 In situ X-ray diffraction probing ferroelectric domains.** In situ X-ray diffraction probing ferroelectric domains: **a** Examples of reciprocal space maps obtained by in situ X-ray near (0 0 1) peak after 10 nm BTO films were grown. Color scales in these maps correspond to linear intensity in arbitrary units. In the image in top left corner we plot the polarization of PTO ultrathin films calculated via Landau theory against PTO thickness and growth temperature. The red circles correspond to the samples grown in this experiment and the displayed reciprocal space maps. The effect of polarization on domains evolution can be seen either in temperature or PTO thickness direction. **b** Evolution of domain sizes plotted against BTO growth time. The domain sizes of BTO films grown on ferroelectric PTO (blue and black data) reach a stable value in the early stage of growth, whereas the ones grown on paraelectric PTO (red and orange data) experience a marked evolution. The domain structures of ultrathin PTO films tend to carry over to BTO film if grown on ferroelectric PTO, which makes the domain evolution more moderate. Source data are provided as a Source Data file.

reciprocal space maps can be assembled into continuous movies that allow the observation of the relaxation process during the growth (Supplementary Movies 1, 2).

A role for polarization in influencing strain relaxation can also be inferred from Landau theory. We again impose the initial boundary condition for BTO growing on PTO that the

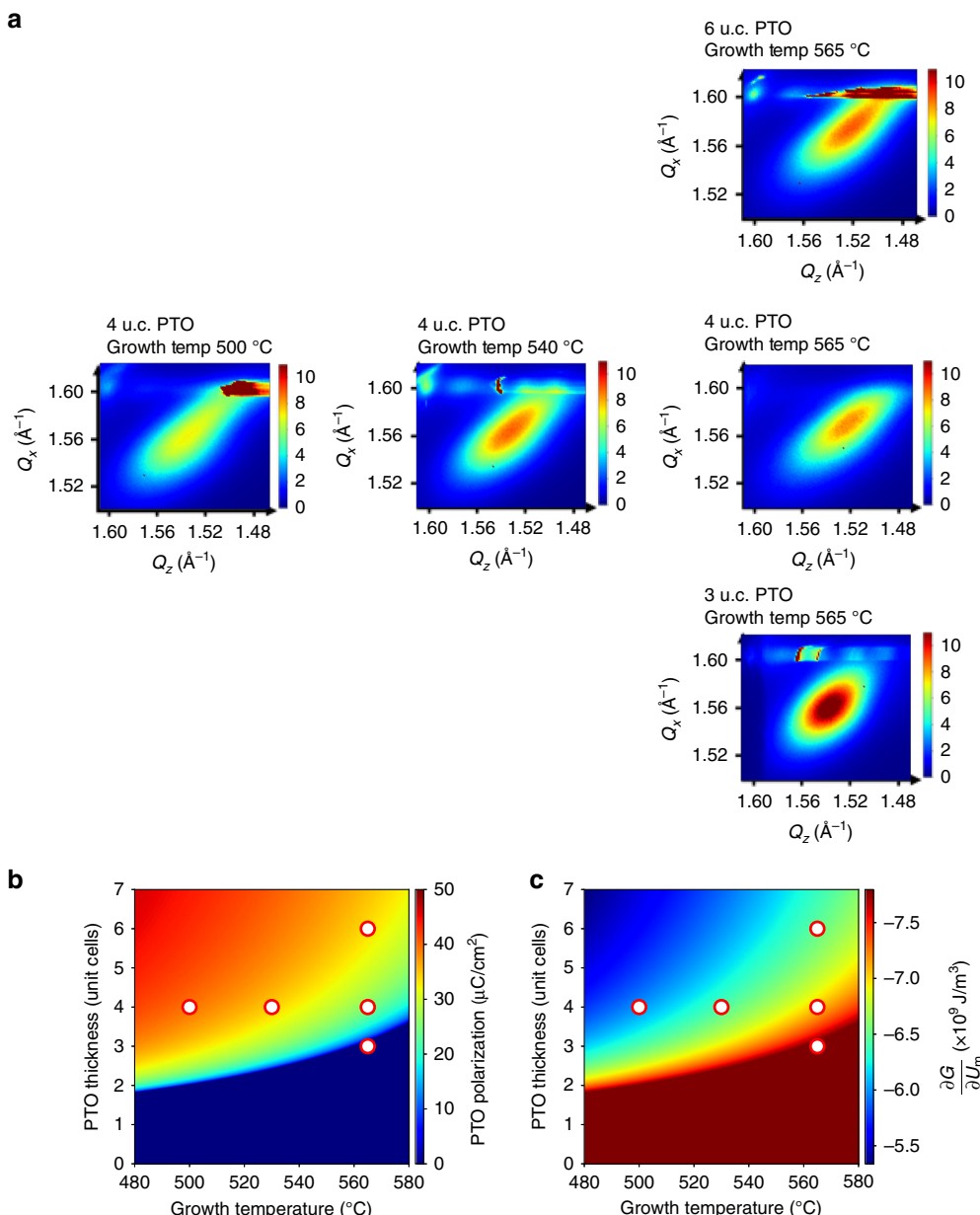

**Fig. 5 In situ x-ray diffraction probing strain relaxation.** In situ x-ray diffraction probing strain relaxation: **a** examples of reciprocal space maps obtained by in situ X-ray near (1 0 1) peak: reciprocal space map images around (1 0 1) peak after 10nm BTO is grown on top of different thickness of PTO grown at different temperature. Color scales in these maps correspond to linear intensity in arbitrary units. **b** Polarization of ultrathin PTO films and **c** $\frac{\partial G}{\partial u_\mathrm{m}}$ of BTO films calculated via Landau theory plotted against thicknesses and temperatures. The red circles in **b** and **c** correspond to the samples in **a**.

polarization is continuous across the interface, take the calculated polarization for the thin PTO layer $P_\mathrm{PTO}$ as the polarization of BTO and calculate $\frac{\partial G}{\partial u_\mathrm{m}}$ for BaTiO₃.

$$\frac{\partial G}{\partial u_\mathrm{m}} = -\frac{2Q_{12}}{s_{11} + s_{12}} P_\mathrm{PTO}^2 + \frac{2u_\mathrm{m}}{s_{11} + s_{12}} \quad (2)$$

In the above equation, the elastic constants are those of BTO, whereas $P_\mathrm{PTO}$ is calculated from Eq. (1) by using the appropriate coefficients for PTO. Under all of the experimental conditions $\frac{\partial G}{\partial u_\mathrm{m}}$ is negative, implying a driving force towards positive strain, i.e., relaxation of the negative misfit strain induced by the substrate. However, it is indeed seen that the polarization of the PTO layer reduces this driving force by making the term $\frac{\partial G}{\partial u_\mathrm{m}}$ more positive.

$\frac{\partial G}{\partial u_\mathrm{m}}$ is plotted as function of PTO thickness and growth temperature in Fig. 5c.

The lattice parameters of relaxed BTO obtained from the center of relaxed parts are plotted against growth time for two representative samples in Fig. 6a. All the films initially grow constrained in-plane to the substrate. Once relaxation begins, the a and c lattice parameters start to evolve towards each other and eventually match bulk BTO lattice parameters. The BTO films grown on PTO with larger polarization stay in the strained ferroelectric state longer and starts relaxation later, but arrives to bulk lattice parameters earlier, implying a quicker relaxation process once it begins. The areas of the relaxed BTO were calculated by adding up all pixels above a certain limit in the relaxed region and are plotted against BTO thickness in (Fig. 6b).

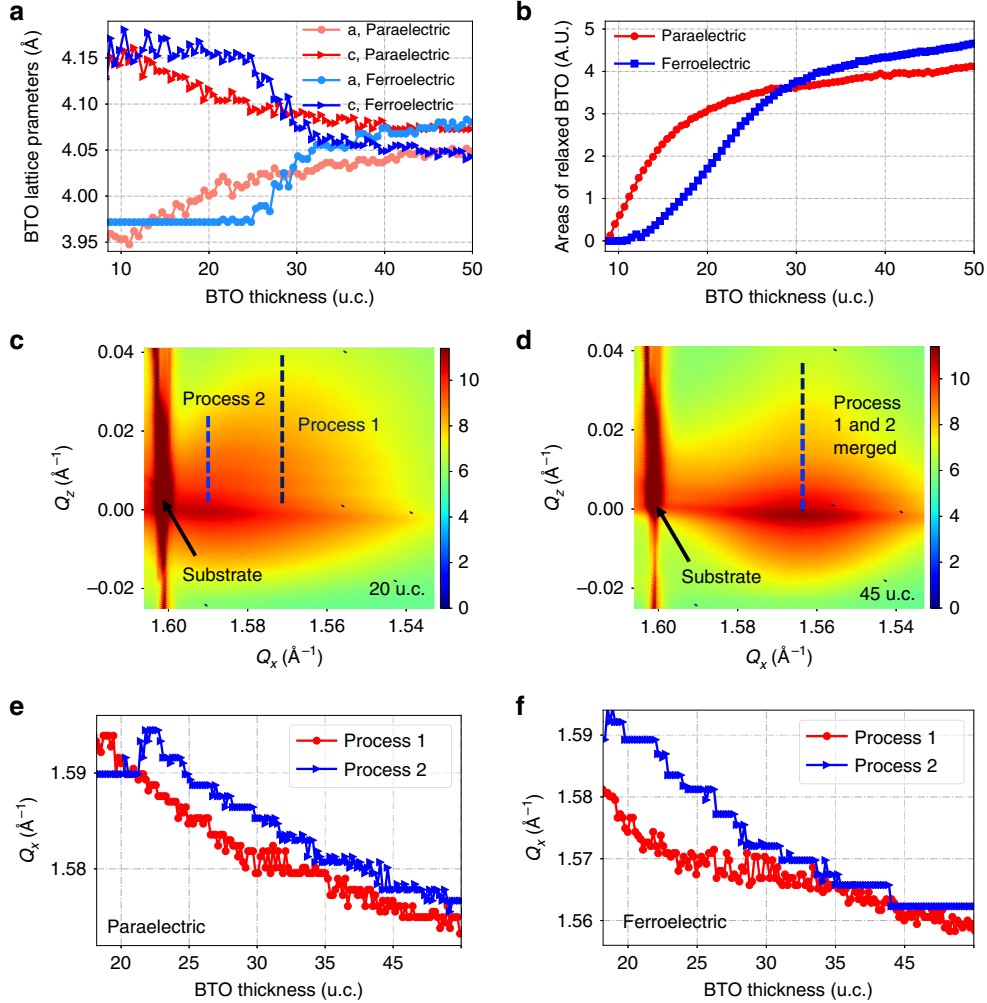

**Fig. 6 Relaxation process. a–b** Relaxation process: **a** In-plane and out-of-plane lattice parameters of relaxed BTO plotted against BTO growth time for two samples: one grown on paraelectric PTO (3 u.c. PTO, 565 °C, red and light red represent lattice parameters **c** and **a**, respectively), and another grown on ferroelectric PTO with largest polarization among all samples (4 u.c. PTO, 500 °C, blue and light blue represent lattice parameters **c** and **a**, respectively). **b** Areas of the relaxed part of BTO film plotted against BTO growth time for two samples described in **a**: red grown on paraelectric PTO and blue grown on ferroelectric PTO. **c, d** Examples of reciprocal space maps obtained by in situ grazing incidence diffraction in the vicinity of the (1 0 0) substrate peak (seen on the left hand side of the map at $Q_x$ ~ 1.60) for BTO films grown on six layers of PTO films at 565 °C after **a** 20 u.c. layers and **b** 45 u.c. layers BTO were grown. Two types of relaxation process were seen, which begin at different $Q_x$ positions in the beginning of growth and merge to the same $Q_x$ positions at the end of the growth. **e, f** Evolution of the $Q_x$ positions of the two relaxation features plotted against BTO thickness for two samples: one grown on paraelectric PTO (**c** 3 u.c. PTO, 580 °C), and another grown on ferroelectric PTO with largest polarization among all samples (**d** 3 u.c. PTO, 500 °C). Red and blue data represent Process 1 and Process 2, respectively. Source data are provided as a Source Data file.

Two different relaxation process of films grown on ferroelectric/paraelectric substrates can be seen.

Further insight into the difference in the in-plane relaxation process can be obtained by performing grazing incidence X-ray scattering around the (1 0 0) peak. Examples from the beginning and end of growth for sample grown on a ferroelectric substrate are shown in Fig. 6c, d. A full set of images for the growth of a sample on a ferroelectric substrate and a paraelectric substrate are shown in Supplementary Figs. 9 and 10 respectively. Two relaxation processes are evident in this data. Process 1 has a larger spread in $Q_z$, whereas Process 2 has a much smaller spread in $Q_z$. On ferroelectric PTO (Fig. 6f) these two process occur at different $Q_x$ positions at the beginning of growth and merge to the same $Q_x$ positions at the end of the growth (this also can be seen by comparing c and d). By contrast the two processes are parallel from the beginning to the end of the growth if the film is grown on paraelectric PTO (Fig. 6e). Process 2 appears to occur through jumps between intermediate states where $Q_x$ remains

constant for a while during relaxation (blue curve in e and f). These jumps are larger and happen less frequently in the case of samples grown on ferroelectric substrates, which is compatible with our argument that the polarization essentially reduces the free energy benefit of strain relaxation and thereby increases the activation energy required for relaxation events to occur.

## Discussion

We can briefly consider scenarios where our findings may be useful. If we consider a device such as a ferroelectric memory based on a capacitor or ferroelectric field effect transistor, films below 10 nm in thickness are not particularly useful owing to increased leakage currents[36] and suppressed polarization owing to depolarization fields[5]. On the other hand, the relaxation of strain as thickness increases leads to lower useful polarization in relaxed films. At a thickness of 10 nm (~25 unit cell layers) a film grown on a paraelectric substrate has lattice parameters near to bulk, whereas one grown on a ferroelectric substrate is completely

strained (Fig. 6a), showing that polarization during growth could have real implications at a thickness of practical importance. In other applications, such as domain wall electronics, where the conductive properties of domain walls can be used as the basis of nanoelectronics[37–43] the density and arrangement of as-grown domains is important, and this also can be modified. As the polarization during growth is a parameter that can be easily and widely adjusted via growth conditions such as temperature and thickness of the underlying ferroelectric layer, there is additional flexibility compared with strain engineering, which is limited by the availability of appropriate substrates. Ferroelectric properties, domain configuration, and the strain state of the thin film can all be manipulated via the underlying ferroelectric polarization of the PTO film during growth, making polarization engineering a powerful approach to the design of tailored ferroelectric films for specialized applications.

## Methods

**Synthesis of films using off-axis RF magnetron sputtering.** Bilayers films of BTO/PTO were grown on 20 nm $SrRuO_3/SrTiO_3$ substrates via off-axis RF magnetron sputtering. The $SrRuO_3$ electrodes were grown at a pressure of 0.1 Torr, Ar: O of 16:3 and a growth temperature of 610 °C. The growth conditions used for the deposition of PTO and BTO were exactly the same: a pressure of 0.18 Torr, Ar:O ratio of 16:7 and the same temperature was used for both films. The growth temperature of the samples spanned the range from 475 °C to 580 °C. The bottom 20 nm thick $SrRuO_3$ electrodes were grown prior to the in situ experiments in the off-axis magnetron sputtering chamber at Stony Brook University. The $SrRuO_3$ electrodes were also atomically flat with single unit cell steps which were checked by AFM. X-ray diffraction results prior to deposition show that these films were epitaxially constrained to the $SrTiO_3$ substrates and had the same in-plane lattice parameter as the substrates.

**Growth rate calibration using fitting method.** The growth time per BTO layer was determined by fitting the $\theta$–$2\theta$ scans and the low-angle reflectivity scans using the same fitting method as in ref. [16]. We can determine the total thickness of film by fitting the low-angle reflectivity scans, and the thickness of each material (with different tetragonality) by fitting the $\theta$–$2\theta$ scans around (0 0 1) and (0 0 2) Bragg peak. All these fitted results confirmed each other to obtain more accurate growth rates. The growth rates at 4-ID beamline was also determined by in situ measurement at the anti-Bragg peak (Supplementary Fig. 6).

**Atomic force and piezoelectric force microscopy.** The films were characterized using an AFM (MFP-3D, Asylum Research) to access the morphology. Vector PFM was conducted to study the ferroelectric domain structure both in-plane and out-of-plane simultaneously. For PFM measurements, conductive Co/Cr-coated silicon tips (spring constant 2.8 N/m) was used and an alternating 0–5 V voltage was applied between the tip and the $SrRuO_3$ bottom electrode.

**Details about in situ x-ray diffraction set-up.** The in situ x-ray diffraction experiments presented here were performed at the NSLS-II 4-ID beamline. At beamline 4-ID, a 2.8 m long undulator is the source of photons, and a monochromator with Si(111) crystals selects ~$2 \times 10^{12}$ photons per second at a photon energy of 11.42 keV with an energy bandwidth of ~$2 \times 10^{4}$ from that source. Mirrors located approximately midway down the beamline with a Pd coating are used to refocus the photons onto the surface of the sample. Beam stability is enhanced with a feedback loop consisting of a diamond beam position detector and piezo actuator for angular adjustment of the second monochromator crystal. The experiments were performed in an in situ growth chamber with temperature, pressure and atmospheric control. Four angles $\phi$, $\theta$, $\delta$, and $2\theta$ can be controlled by computer during the experiments. The x-ray beam enters the chamber through a beryllium window, scatters off the sample, exits via a second beryllium window and is detected by an Eiger 1M area detector. In the grazing incidence experiment, the sample is inclined by a small grazing angle with respect to the incoming X-ray beam.

**In situ x-ray diffraction data analysis.** The in situ x-ray diffraction data obtained at 4-ID beamline were saved in HDF5 files. A methodology was developed using Python to analyze the files. The 2D array data were converted to 3D data with angles and reciprocal space parameters to perform reciprocal space maps. The reciprocal space growth movies were generated by merging all the maps in the order of time. The x-ray intensity of different samples were normalized to be comparable. The relaxed area of BTO films were defined by setting up a certain threshold for all samples after normalization. The positions of multipeaks were obtained using several methods to reduce errors.

## Data availability
The source data underlying Figs. 1, 3a, 4b, 6a, b, e, f and Supplementary Figs. 1, 3, 4, 5, 7 are provided as a Source Data file. The raw data files for these experiments very large and can be accessed by request to the authors.

## Code availability
No significant custom code was developed for this work. Data from NSLS-II were processed using beamline software available to all users of the facility. Theoretical calculations were made using commercially available Maple software.

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

## Acknowledgements

This work was supported by NSF DMR-1055413, DMR-1334867, and DMR-1506930. Development of the in situ growth facility used in this work was supported by NSF DMR-0959486. This research used beamline 4-ID of the National Synchrotron Light Source II, an US Department of Energy (DOE) Office of Science User Facility operated for the DOE Office of Science by Brookhaven National Laboratory under contract no. DE-SC0012704. We thank C. Nelson and Z. Yin for assistance with the experiments at 4-ID.

## Author contributions

R.L., J.G.U., H.-C.H, A.G., B.B, G.B., A.L., K.E.-L., R.L.H., and M.D participated in the experimental work at NSLS-II. R.L. conducted the bulk of the experimental work outside of NSLS-II and supervised A.S., C.P., K.C., and E.D. who completed high school and undergraduate research projects related to the research. R.L. was chiefly responsible for processing and analysis of the data. The experimental apparatus was designed, assembled, and integrated in to the 4-ID beamline by R.L.H. and M.D. M.D. devised the experimental methods and directed the project. The manuscript was prepared by R.L. and M.D. with contributions from the other authors.

## Competing interests

The authors declare no competing interests.
