## [Peer Review File · Nature Communications]

Reviewers' comments:

Reviewer #1 (Remarks to the Author):

The manuscript "Role of ferroelectric polarization during growth of highly strained ferroelectrics revealed by in-situ x-ray diffraction" by Rui Liu et al. presents a significant contribution to our fundamental understanding of the role of electric polarization in the growth process of epitaxial thin films. This work helps to fill a gap in studies of complex oxide multilayer growth and properties as the contribution of polarization coupling effects is often overlooked.

The central claim of the manuscript is that the electric polarization state of an oxide layer (lead titanate, PTO) affects the growth rate, relaxation, and polarization domain structure of the layer (barium titanate, BTO) that is grown on top of PTO. The ferroelectric polarization state of PTO layers is controlled by PTO film thickness and temperature before BTO layer deposition. As coincidence does not imply causation, it has been important for the authors to present strong arguments that the observed changes in structure and properties of BTO films are caused by PTO polarization and not by temperature or strain coupling. I think that experimental data presented along with thermodynamic theory modeling are sufficient to support the main claim of the article. The weaknesses of the manuscript in my opinion are that (1) the work is narrowly focused on one system only with little or no discussion of the applicability of findings for other systems and materials and (2) modeling does not include electrostatic interactions between materials explicitly.

I believe this work can be important and of great interest to a broad range of fundamental science and materials engineering researchers. I can recommend this manuscript for publication in Nature Communications provided the authors consider the following questions and comments:

1) The manuscript in its present form is very narrowly focused on one system, BTO/PTO, and broader impacts of this research are briefly mentioned only at the very end of the article. I think it is important to consider other ferroelectric/ferroelectric systems in which the authors expect to find similar polarization-controlled growth effects. Also, I suggest the significance of this research to broader audience in the field of thin-film research is emphasized more at the beginning of the paper (and/or in the abstract).

2) What is the origin of polarization-controlled growth? Generally, the film growth depends on the surface and interface energies. It could be useful to estimate the changes of the surface energies with polarization (and temperature) instead of using the Gibbs free energy.

3) Can we expect to find similar polarization-controlled growth effects in multilayers including non-ferroelectric layers? STO on PTO? ZnO on PTO?

4) The authors assume for thermodynamic theory calculations that the spontaneous polarization of the PTO layer can be imposed on BTO and used to analyze the stability of the BTO layer that grows on PTO. In fact, it is more likely that polarizations in both BTO and PTO layers will be different from their thermodynamic equilibrium values of pure substances due to electrostatic polarization coupling. The total free energy of the system will be the sum of $G(\text{BTO})$, $G(\text{PTO})$, and an extra energy term due to electrostatic polarization coupling (see, for example, Okatan et al. PRB 79, 174113 (2009)). Ideally, and this is especially important when the film thickness increases, the authors should consider adding coupling to their calculations and present the total free energy of the multilayer system instead of the energies of individual non-interacting layers.

5) It has been predicted that the electrostatic coupling is weaker in thicker ferroelectric layers. For instance, a transition from strong to weak coupling in BTO/PZT (Salev et al. PRB 93, 041423R (2016)) in the range of film thicknesses between 10 nm and 30 nm leads to different polarizations (close to single-layer polarization values) in different layers. Could it be that the BTO layer relaxation transition in films thicker than 25 u.c. (Fig. 6 in the manuscript) is related to the

coupling transition that can only occur when the substrate (PTO) is ferroelectric?

Reviewer #2 (Remarks to the Author):

This is a very interesting paper that describes experiments demonstrating that the ferroelectric polarization of thin PbTiO₃ layers can have a significant effect on the growth behavior and subsequent room T properties of BaTiO₃. The authors demonstrate convincingly that the polarization magnitude of the PbTiO₃ layer is responsible for this interesting behavior. Although I see one problem with the paper (described in the following), overall I feel that the study presents interesting and potentially important information of broad general interest. Thus, I strongly support publication of the paper in Nature Communications after editing to address the following issue.

The one concern I have is that the model described starting in the bottom paragraph of page 9 indicates that stripe domains in the PbTiO₃ layer play a significant role in controlling the domain structure of the BaTiO₃. They cite a paper by Fong et al. (ref 9) as evidence that the thin PTO layers form 180° stripe domains. However, it should be noted that the results described in ref. 9 were for films grown directly on SrTiO₃ substrates. Subsequent studies by that same group found that films grown on conducting SrRuO₃ layers on STO are monodomain, since the SRO layer provides electrical compensation at the lower PTO interface (see Fong et al. PRL 96, 127601 (2006)). Since the PTO layers in the current study were grown on conducting SRO layers, they are very unlikely to have 180° stripe domains. The authors need to modify the manuscript to take this into account.

Reviewer #3 (Remarks to the Author):

Review of Liu et al,

"Role of ferroelectric polarization during growth of highly strained ferroelectrics revealed by in-situ x-ray diffraction"

The primary result of this paper is the intriguing observation that if BaTiO₃ (BTO) is sputter-deposited on a polarized interface, the growth rate unexpectedly is lowered in a systematic way based on the polarization strength at the surface. If the surface is tuned to be non-polar, then the growth rate is constant as expected. As noted by the authors, this points towards exciting additional ways to engineer ferroelectric thin films materials properties during growth.

I agree that this is an interesting and novel observation and would be of interest to others in the community, although I apologize in advance to the authors that I do not recommend publication of the paper in its current form.

The authors performed a set of experiments that tuned the polarization strength at the surface by using ferroelectric properties of specially prepared heterostructure substrates created with a ultrathin layers of coherently strained PbTiO₃ (PTO) on top of 20nm SrTiO₃ epitaxial on top of SrTiO₃ (100) crystal substrate. At a range of temperature around the typical growth temperatures for the BTO, the PTO undergoes a phase transition from paraelectric or ferroelectric, and the magnitude of the polarization in the ferroelectric phase depends on temperature and thickness of the underlying PTO film.

The authors show a plausible and reasoned argument that suggests this unanticipated effect is not due to details of sputter deposition process (such as the changing proportion of neutral/ionized BTO subspecies with sputtering conditions) but might be explained using a thermodynamic argument based attachment of BTO unit cell (uc) if it is required to be in a polarized versus un-

polarized state.

In addition, there are several secondary complications which they also documented during their experiments. These are all related to the significant strain relaxation unavoidably occurring during their depositions as the BTO film becomes thicker than about 10-15 nm. I note that strain relaxation could be expected to have an effect on growth rate, although one might assume that deposition is 'easier' when the film is relaxed closer to the bulk lattice parameters than when species are trying to maintain a coherently strained lattice. If this had an effect on growth rate, it would be opposite to what is observed in this paper.

They observe that the strain relaxation in the BTO film growth is suppressed somewhat for a BTO film grown on a polarized surface, coincident with the growth mode of BTO presenting a more smooth or layer-by-layer aspect on the polarized surface, but the growth mode is rather rough on the un-polarized surface. (See Supplemental Figure 6 and 7) The authors note that these two aspects are correlated and also suggest that the PTO polarization (which is expected to enhance tetragonality in the BTO) also helps the BTO maintain coherency for slightly thicker films.

I am very interested in the results and their interpretations.

Unfortunately, while I believe the authors have a careful and systematic set of experiments and have thoughtfully combined many techniques and supporting characterizations that may provide a complete picture to themselves, I also had a overly hard time parsing through this paper due to its limitations.

The presentation of the work is disorganized and in places, sloppy. I do not recommend the publication of this work in its current form. I suggest that the authors take the time to tighten up the organization, check their statements for precision and clarity, check their figures and descriptions of their methods for accuracy.

While the following is not exhaustive, I have outlined some of the issues that I puzzled over. If I have misinterpreted some of the experiments, that is consistent with the issues I had in working with this paper in its current form.

A) What are the implications or issues of measuring a growth rate that may be varying over time?:

Fig 1 shows the primary proof of the assertion of the BTO growth rate decreases with increasing surface polarization. As far as I can tell from the paper, I assume that the growth rates must have been (?) determined by sputter depositing an approximately 20nm BTO film, then knowing the time and measuring the thickness via XRD (Supplemental Fig 1) gives the growth rate.

This protocol would give the *average growth rate* over 50 unit cells of the 20 nm BTO film (from coherency to partially or fully relaxed). Note that different systems shown spend different amounts of time in various stages of relaxation that depends on the thickness of PTO underlayer and the growth temperature.

Consider the comparison growth 'rate' for the 50nm BTO/PTO superlattice on SRO(?) /STO. I would assume that the 1st 3 unit cells of BTO grown on the 1st 3 unit cells of PTO on the 'virgin' SRO/STO substrate, should act *identically* (SAME GROWTH RATE) to the 1st 3 unit cells of the 20nm BTO film grown on a 3uc PTO/SRO/STO substrate. The average growth rate, however, as documented in the figure, ends up completely different.

In the subsequent synchrotron experiments, the authors note that they measured the growth rate from the CTR oscillations at the BTO (0 0 1/2) position (Supplemental Fig 6). This could distinguish the rate for the 1st few unit cells (before significant relaxation) and isolate it from an average.

Were the in-situ synchrotron xray measurements consistent with the growth rates and trends shown in Figure 1? In particular, what are the rates for the first few BTO unit cells before relaxation complicates things?

It is possible that these calibration growths were only done for a few samples (in order to set the sputtering conditions to get roughly the same growth rate in the new chamber as the in-lab experiments?). Perhaps the desired data doesn't exist, but that wasn't clear from the descriptions of methods.

B) Supplemental Figure 1 - the x-axis is not labeled (sloppy):

It is noted that the plot is from theta/2theta XRD scan, but is the x-axis listing the theta or twotheta values? Also why not label some of the features - I assume the peak is a resolution broadened STO(?) 001. I calculated fringe oscillations on this curve that the SRO and the BTO films are each closer to 10nm rather than the 20nm stated, but I could be mistaken.

C) Inconsistency in domain size in AFM images and the x-ray results (micron compared to 30nm domains):

Figure 3 shows piezo-AFM, with ferroelectric domain sizes being micron or sub micron. Diffuse scattering in Figure 4 shows domain sizes in these films should be approximately 20-60nm, or basically about 30nm at end of growth.

1) Why are there only sub-micron/micron size domains in the p-AFM and no sign of the finer domain structure?

2) If the domains coarsen that significantly when the samples are cooled, is this consistent with the x-ray results? (as the samples are cooled further, if the domains coarsen, the diffuse scattering would crawl back under the CTR. Was this observed?)

D) Inconsistency in the calculation of the domain size in Figure 4:

Although the images are highly saturated in order to help show the diffuse scattering, in Figure 4a) the 3uc 300C image shows clear maxima in the diffuse scattering at about 0.01 inverse angstroms from the CTR. This is a periodicity of about 60 nm (600 angstroms = $2\pi/0.01$). Assuming diffuse peak is from correlations from ~ 1 up/ ~ 1 down domain periodicity, then each up or down domain itself about 30nm. These images are on 10nm thick films, or about 25 unit cells. However, In Figure 4b, it suggests the automatic fitting procedure of the domains gave a domain size of 25nm which seems low compared to the image even despite the graphical estimations I did on the figures.

To help with showing more features in the colormap, plot $\log(I)$ instead of Intensity. If intensity colormap is $\log(I)$, then that should be noted explicitly, right?

E) Inconsistency in Supplemental Figure 7 (sloppy?):

In supplemental figure 7a and b, the curves that are shown by the red dots in a and b should be identical since they are the identical sample [4uc PTO at 540C]. They are not identical. Is this in error? If these are nominally identically prepared samples but two different experimental runs, then it also gives some indication of reproducibility and should be taken into account in other aspects of the work.

F) (unclear or rationale unclear):

In the preparation of the underlying heterostructures (n uc PTO/20nm SRO/STO(100) it was implied that each ultrathin layer of PTO was grown at different temperatures (based on its calculated T_c ?) ("ultrathin PTO films were grown at temperatures in the vicinity of the ferroelectric phase transition temperature"), which would be a very odd temperature to grow the PTO underlayer rather than creating identically grown heterostructures and only varying the subsequent temperatures in the BTO growths, however, perhaps I didn't understand the rationale or misinterpreted the statement.

Wouldn't using similar temperature and conditions be rather critical in creating as similar a surface

morphology of the ultrathin PTO?

G) Supplemental Figure 1 as shown adds to reader confusions rather than illuminates:

This figure showing the CTR's for 3 samples grown to approximately 20nm. There is no significant difference in total final thickness and thus growth rate is roughly constant, despite the fact that they cover the three temperatures below/at/above the calculated TC of the 3uc PTO underlayer substrate. Initially this figure confused me as being inconsistent with the authors claims that the phase transition would affect the rate. Then I noticed that Figure 1 in the paper similarly shows NO significant change in growth rate for this particular sample at those particular temperatures).

Wouldn't it be useful to add one further temperature for comparison or show different sample that actually has a growth rate difference to help reader visualize the experiments better.

Or explain to reader more clearly what they are supposed to be learning or visualizing from this figure.

Reviewers' comments:

We thank all of the reviewers for their constructive comments and have modified the paper significantly in response to them. Point by point responses can be found below. Substantive changes made to the manuscript are in red text in the revised version, as are our point by point responses in this document.

Reviewer #1 (Remarks to the Author):

The manuscript "Role of ferroelectric polarization during growth of highly strained ferroelectrics revealed by in-situ x-ray diffraction" by Rui Liu et al. presents a significant contribution to our fundamental understanding of the role of electric polarization in the growth process of epitaxial thin films. This work helps to fill a gap in studies of complex oxide multilayer growth and properties as the contribution of polarization coupling effects is often overlooked.

The central claim of the manuscript is that the electric polarization state of an oxide layer (lead titanate, PTO) affects the growth rate, relaxation, and polarization domain structure of the layer (barium titanate, BTO) that is grown on top of PTO. The ferroelectric polarization state of PTO layers is controlled by PTO film thickness and temperature before BTO layer deposition. As coincidence does not imply causation, it has been important for the authors to present strong arguments that the observed changes in structure and properties of BTO films are caused by PTO polarization and not by temperature or strain coupling. I think that experimental data presented along with thermodynamic theory modeling are sufficient to support the main claim of the article. The weaknesses of the manuscript in my opinion are that (1) the work is narrowly focused on one system only with little or no discussion of the applicability of findings for other systems and materials and (2) modeling does not include electrostatic interactions between materials explicitly.

I believe this work can be important and of great interest to a broad range of fundamental science and materials engineering researchers. I can recommend this manuscript for publication in Nature Communications provided the authors consider the following questions and comments:

1) The manuscript in its present form is very narrowly focused on one system, BTO/PTO, and broader impacts of this research are briefly mentioned only at the very end of the article. I think it is important to consider other ferroelectric/ferroelectric systems in which the authors expect to find similar polarization-controlled growth effects. Also, I suggest the significance of this research to broader audience in the field of thin-film research is emphasized more at the beginning of the paper (and/or in the abstract).

We have added some discussion of where we expect this behavior to be observed both in the abstract and the main body of the paper. This broadens the significance of the research and we thank the referee for their suggestion.

2) What is the origin of polarization-controlled growth? Generally, the film growth depends on the surface and interface energies. It could be useful to estimate the changes of the surface energies with polarization (and temperature) instead of using the Gibbs free energy.

The model we have advanced suggests that in this case the change in the growth is driven by the bulk characteristics of the layer being grown. We agree that modelling of the surface and interface energies would be an interesting approach, but hope that our primarily experimental paper will inspire theorists to undertake this work, and for ourselves prefer to maintain our model, which despite being very oversimplified, accounts well for the behavior observed. We have however expanded our discussion of the modelling approach to discuss why the bulk energy becomes important in the present case and given some guidance for when we expect it to be relevant in other similar systems.

3) Can we expect to find similar polarization-controlled growth effects in multilayers including non-ferroelectric layers? STO on PTO? ZnO on PTO?

In principle we might expect these effects in multilayers with non-ferroelectric layers. However in the case of STO on PTO it turns out that the large dielectric constant of STO means that in practice these effects are not seen, consistent with our experimental observations. We have added a discussion on this point to the revised paper. One might indeed expect to see the effect in ZnO on PTO, but we have no experience growing this system.

4) The authors assume for thermodynamic theory calculations that the spontaneous polarization of the PTO layer can be imposed on BTO and used to analyze the stability of the BTO layer that grows on PTO. In fact, it is more likely that polarizations in both BTO and PTO layers will be different from their thermodynamic equilibrium values of pure substances due to electrostatic polarization coupling. The total free energy of the system will be the sum of $G(\text{BTO})$, $G(\text{PTO})$, and an extra energy term due to electrostatic polarization coupling (see, for example, Okatan et al. PRB 79,

174113 (2009)). Ideally, and this is especially important when the film thickness increases, the authors should consider adding coupling to their calculations and present the total free energy of the multilayer system instead of the energies of individual non-interacting layers.

While we agree that for thicker layers it would be appropriate to include a coupling term we wanted to keep our model as simple as possible and capture only the initial energetics as the BTO layer starts to grow. Here the difference between the polarization is minimal and this term can be neglected. It would definitely be interesting to model the full growth process, which would certainly require this term to be included, but we feel this is beyond the scope of the present work. We have added some discussion on this point to the paper and cited the reference mentioned by the referee.

5) It has been predicted that the electrostatic coupling is weaker in thicker ferroelectric layers. For instance, a transition from strong to weak coupling in BTO/PZT (Salev et al. PRB 93, 041423R (2016)) in the range of film thicknesses between 10 nm and 30 nm leads to different polarizations (close to single-layer polarization values) in different layers. Could it be that the BTO layer relaxation transition in films thicker than 25 u.c. (Fig. 6 in the manuscript) is related to the coupling transition that can only occur when the substrate (PTO) is ferroelectric?

This is an interesting suggestion, however we believe that the relaxation of the BTO is simply due to the inevitable desire to release the imposed strain, which can be postponed by the PTO polarization but not indefinitely. While a change in polarization coupling could potentially occur as the film grows we would prefer not to speculate about that without having experimental results that would more directly support this suggestion.

Reviewer #2 (Remarks to the Author):

This is a very interesting paper that describes experiments demonstrating that the ferroelectric polarization of thin PbTiO₃ layers can have a significant effect on the growth behavior and subsequent room T properties of BaTiO₃. The authors demonstrate convincingly that the polarization magnitude of the PbTiO₃ layer is responsible for this interesting behavior. Although I see one problem with the paper (described in the following), overall I feel that the study presents interesting and potentially important information of broad general interest. Thus, I strongly support publication of the paper in Nature Communications after editing to address the following issue.

The one concern I have is that the model described starting in the bottom paragraph of page 9 indicates that stripe domains in the PbTiO₃ layer play a significant role in controlling the domain structure of the BaTiO₃. They cite a paper by Fong et al. (ref 9) as evidence that the thin PTO layers form 180° stripe domains. However, it should be noted that the results described in ref. 9 were for films grown directly on SrTiO₃ substrates. Subsequent studies by that same group found that films grown on conducting SrRuO₃ layers on STO are monodomain, since the SRO layer provides electrical compensation at the lower PTO interface (see Fong et al. PRL 96, 127601 (2006)). Since the PTO layers in the current study were grown on conducting SRO layers, they are very unlikely to have 180° stripe domains. The authors need to modify the manuscript to take this into account.

We thank the referee for their positive remarks. We have modified our description of the results of Fong et al to reflect the findings in both papers.

Reviewer #3 (Remarks to the Author):

Review of Liu et al,

"Role of ferroelectric polarization during growth of highly strained ferroelectrics revealed by in-situ x-ray diffraction"

The primary result of this paper is the intriguing observation that if BaTiO₃ (BTO) is sputter-deposited on a polarized interface, the growth rate unexpectedly is lowered in a systematic way based on the polarization strength at the surface. If the surface is tuned to be non-polar, then the growth rate is constant as expected. As noted by the authors, this points towards exciting additional ways to engineer ferroelectric thin films materials properties during growth.

I agree that this is an interesting and novel observation and would be of interest to others in the community, although I apologize in advance to the authors that I do not recommend publication of the paper in its current form.

The authors performed a set of experiments that tuned the polarization strength at the surface by using ferroelectric properties of specially prepared heterostructure substrates created with a ultrathin layers of coherently strained PbTiO₃ (PTO) on top of 20nm SrTiO₃ epitaxial on top of SrTiO₃ (100) crystal substrate. At a range of temperature around the typical growth temperatures for the BTO, the PTO undergoes a phase transition from paraelectric or

ferroelectric, and the magnitude of the polarization in the ferroelectric phase depends on temperature and thickness of the underlying PTO film.

The authors show a plausible and reasoned argument that suggests this unanticipated effect is not due to details of sputter deposition process (such as the changing proportion of neutral/ionized BTO subspecies with sputtering conditions) but might be explained using a thermodynamic argument based attachment of BTO unit cell (uc) if it is required to be in a polarized versus un-polarized state.

In addition, there are several secondary complications which they also documented during their experiments. These are all related to the significant strain relaxation unavoidably occurring during their depositions as the BTO film becomes thicker than about 10-15 nm. I note that strain relaxation could be expected to have an effect on growth rate, although one might assume that deposition is 'easier' when the film is relaxed closer to the bulk lattice parameters than when species are trying to maintain a coherently strained lattice. If this had an effect on growth rate, it would be opposite to what is observed in this paper.

They observe that the strain relaxation in the BTO film growth is suppressed somewhat for a BTO film grown on a polarized surface, coincident with the growth mode of BTO presenting a more smooth or layer-by-layer aspect on the polarized surface, but the growth mode is rather rough on the un-polarized surface. (See Supplemental Figure 6 and 7) The authors note that these two aspects are correlated and also suggest that the PTO polarization (which is expected to enhance tetragonality in the BTO) also helps the BTO maintain coherency for slightly thicker films.

I am very interested in the results and their interpretations.

Unfortunately, while I believe the authors have a careful and systematic set of experiments and have thoughtfully combined many techniques and supporting characterizations that may provide a complete picture to themselves, I also had a overly hard time parsing through this paper due to its limitations.

The presentation of the work is disorganized and in places, sloppy. I do not recommend the publication of this work in its current form. I suggest that the authors take the time to tighten up the organization, check their statements for precision and clarity, check their figures and descriptions of their methods for accuracy.

We hope that the reviewer will find the revised version of the manuscript clearer. We did go through and change several statements that were potentially unclear and have addressed all of their specific concerns as outlined below.

While the following is not exhaustive, I have outlined some of the issues that I puzzled over. If I have misinterpreted some of the experiments, that is consistent with the issues I had in working with this paper in its current form.

A) What are the implications or issues of measuring a growth rate that may be varying over time?:

Fig 1 shows the primary proof of the assertion of the BTO growth rate decreases with increasing surface polarization. As far as I can tell from the paper, I assume that the growth rates must have been (?) determined by sputter depositing an approximately 20nm BTO film, then knowing the time and measuring the thickness via XRD (Supplemental Fig 1) gives the growth rate.

This protocol would give the *average growth rate* over 50 unit cells of the 20 nm BTO film (from coherency to partially or fully relaxed). Note that different systems shown spend different amounts of time in various stages of relaxation that depends on the thickness of PTO underlayer and the growth temperature.

Consider the comparison growth 'rate' for the 50nm BTO/PTO superlattice on SRO(?)/STO. I would assume that the 1st 3 unit cells of BTO grown on the 1st 3unit cells of PTO on the 'virgin' SRO/STO substrate), should act *identically* (SAME GROWTH RATE) to the 1st 3 unit cells of the 20nm BTO film grown on a 3uc PTO/SRO/STO substrate. The average growth rate, however, as documented in the figure, ends up completely different.

As the referee note the growth rates were determined by knowing the sputtered time and fitted the thickness via XRD (Both 2theta-omega and reflectivity scans), which gives the *average growth rate* over 50 unit cells of the 20 nm BTO film. The difference that the referee noticed between superlattices and bilayers is an excellent point and we have added discussion about it.

In the subsequent synchrotron experiments, the authors note that they measured the growth rate from the CTR oscillations at the BTO (0 0 1/2) position (Supplemental Fig 6). This could distinguish the rate for the 1st few unit cells (before significant relaxation) and isolate it from an average.

We have followed the referees suggestion and drawn attention to Supplementary Figure 6 when discussing the discrepancy in growth rates between the bilayer and the superlattice.

Were the in-situ synchrotron xray measurements consistent with the growth rates and trends shown in Figure 1? In particular, what are the rates for the first few BTO unit cells before relaxation complicates things?

It is possible that these calibration growths were only done for a few samples (in order to set the sputtering conditions to get roughly the same growth rate in the new chamber as the in-lab experiments?). Perhaps the desired data doesn't exist, but that wasn't clear from the descriptions of methods.

Due to differences in chamber geometry the growth rates are not exactly the same for the in-situ experiments and those performed in our laboratory. However the trends observed are consistent. We did only do a few calibration measurements of growth rates at the synchrotron (these measurements cannot be performed simultaneously with our maps, so to do them for every sample would double the required amount of beamtime). The measurements that were performed did match the change in growth rate as samples are made thicker (ie. The first few unit cells grew faster than subsequent ones) and this is now mentioned in the paper.

B) Supplemental Figure 1 - the x-axis is not labeled (sloppy):

It is noted that the plot is from theta/2theta XRD scan, but is the x-axis listing the theta or twotheta values? Also why not label some of the features - I assume the peak is a resolution broadened STO(?) 001. I calculated fringe oscillations on this curve that the SRO and the BTO films are each closer to 10nm rather than the 20nm stated, but I could be mistaken.

The x-axis displays 2theta values and the thickness of the film is 20nm. It is possible the referee used the labeled values as theta instead of 2theta, which gives half the thickness. We apologize for any confusion.

C) Inconsistency in domain size in AFM images and the x-ray results (micron compared to 30nm domains):

Figure 3 shows piezo-AFM, with ferroelectric domain sizes being micron or sub micron. Diffuse scattering in Figure 4 shows domain sizes in these films should be approximately 20-60nm, or basically about 30nm at end of growth.

1) Why are there only sub-micron/micron size domains in the p-AFM and no sign of the finer domain structure?

The larger micron scale domain structure is that written with the AFM tip. The smaller scale texture in the image is from the naturally occurring domains which are indeed on the 20-60nm length scale. We have added some clarification to the figure caption for Figure 3.

2) If the domains coarsen that significantly when the samples are cooled, is this consistent with the x-ray results? (as the samples are cooled further, if the domains coarsen, the diffuse scattering would crawl back under the CTR. Was this observed?)

We measured domain sizes during cooling and no significant changes happened. We apologize for the confusion between the written domain structure and the small scale naturally occurring domain structure.

D) Inconsistency in the calculation of the domain size in Figure 4:

Although the images are highly saturated in order to help show the diffuse scattering, in Figure 4a) the 3uc 300C image shows clear maxima in the diffuse scattering at about 0.01 inverse angstroms from the CTR. This is a periodicity of about 60 nm (600 angstroms = $2\pi/0.01$). Assuming diffuse peak is from correlations from ~1up/~1down domain periodicity, then each up or down domain itself about 30nm. These images are on 10nm thick films, or about 25 unit cells. However, In Figure 4b, it suggests the automatic fitting procedure of the domains gave a domain size of 25nm which seems low compared to the image even despite the graphical estimations I did on the figures.

The labelling of the axis makes it a little hard to read, but the size of the domains is consistent in Fig 4. Given the uncertainties involved we are actually quite impressed that the analysis the referee made corresponded as closely as it did with our results in which are able to fit the raw data.

To help with showing more features in the colormap, plot $\log(I)$ instead of Intensity. If intensity colormap is $\log(I)$, then that should be noted explicitly, right?

We produced a new version of the colormaps as $\log(I)$, but to our mind it is not a clear improvement. We decided to keep our original version. The $\log(I)$ version is included below for the referee.

E) Inconsistency in Supplemental Figure 7 (sloppy?):

In supplemental figure 7a and b, the curves that are shown by the red dots in a and b should be identical since they are the identical sample [4uc PTO at 540C]. They are not identical. Is this in error? If these are nominally identically prepared samples but two different experimental runs, then it also gives some indication of reproducibility and should be taken into account in other aspects of the work.

In fact the blue data in (a) and the red data in (b) are identical, however in the previous version of the paper the temperature for b and d were incorrect in our caption and we see how the referee was misled by our mistake. We apologize for the mistake and have corrected it.

F) (unclear or rationale unclear):

In the preparation of the underlying heterostructures (n uc PTO/20nm SRO/STO(100)) it was implied that each ultrathin layer of PTO was grown at different temperatures (based on its calculated T_c) ("ultrathin PTO films were grown at temperatures in the vicinity of the ferroelectric phase transition temperature"), which would be a very odd temperature to grow the PTO underlayer rather than creating identically grown heterostructures and only varying the subsequent temperatures in the BTO growths, however, perhaps I didn't understand the rationale or misinterpreted the statement.

Wouldn't using similar temperature and conditions be rather critical in creating as similar a surface morphology of the ultrathin PTO?

Indeed each ultrathin layer of PTO was grown at different temperatures, which gives it the different polarization characteristics. We performed the experiments this way to keep the time delay between the PTO and BTO layer growth consistent. To a certain extent this choice was actually inspired by the origins of this project in growing finely layered PTO/BTO superlattices. In superlattices we have tended to find that it is detrimental to adjust the growth temperature from one layer to the next as the additional time required between layers leads to lower film quality.

G) Supplemental Figure 1 as shown adds to reader confusions rather than illuminates:

This figure showing the CTR's for 3 samples grown to approximately 20nm. There is no significant difference in total final thickness and thus growth rate is roughly constant, despite the fact that they cover the three temperatures below/at/above the calculated TC of the 3uc PTO underlayer substrate. Initially this figure confused me as being inconsistent with the authors claims that the phase transition would affect the rate. Then I noticed that Figure 1 in the paper similarly shows NO significant change in growth rate for this particular sample at those particular temperatures).

Wouldn't it be useful to add one further temperature for comparison or show different sample that actually has a

growth rate difference to help reader visualize the experiments better.

Or explain to reader more clearly what they are supposed to be learning or visualizing from this figure.

We agree that the samples chosen in the figure were poor choices as the growth rate did not change too much so we changed the presented samples to grown at 475C, 550C, 565C. Here 550C and 565C are beyond the transition temperatures of 3 u.c. PTO layer and have similar growth rates, which are identifiably different to that of the sample grown at 475C.

Reviewers' comments:

Reviewer #1 (Remarks to the Author):

I think the latest version of the manuscript has addressed all my suggestions and concerns adequately. Also, the authors improved the presentation and description of the results. I have no further reservations, and I can recommend the paper for publication in its present state.

Reviewer #2 (Remarks to the Author):

While I still feel that, overall, this is a very interesting paper appropriate for publication in Nature Communications, I am not yet satisfied with the authors' response to my initial review. The authors perhaps didn't understand my comment regarding the effect on stripe domain formation of including a SrRuO₃ conducting electrode between the SrTiO₃ substrate and the PTO or PTO/BTO films. The revised version of the paper includes the sentence "Depending on the electrical boundary conditions underneath them, Fong et al [9, 10], found that ultra-thin PTO layers have a stripe domain structure which produce diffraction features." This wording is very misleading (besides having a typo - should be "produces" rather than "produce"). What Fong et al. actually found is that PTO films grown directly on STO form a stripe domain structure, while films grown on a SrRuO₃ conducting bottom electrode DO NOT form a stripe domain structure, but rather are monodomain (other than within a very narrow range of environmental conditions when adjusting the oxygen partial pressure to induce a switch between monodomain up and monodomain down states). While the authors are vague on this point, they seem to be concluding that their BTO films, which are grown on top of monodomain PTO, form 180° stripe domains (they say, for example, "our experimental results suggest the ferroelectric polarization of the PTO film underneath can prevent or delay this relaxation and help the BTO film stay in a strained state, protecting the tetragonal polarization domains."). Why would tetragonal polarization domains form when the growth is on a substrate that includes a conducting SrRuO₃ bottom electrode (which would result in monodomain PTO)? What is the evidence that the BTO isn't monodomain? The diffuse intensity shown in Fig. 4 is not convincing evidence of stripe domains. The diffuse intensity shown in those images is, with only one exception, centered on the CTR, so isn't consistent with the scattering expected for stripe domains. How are the domain sizes shown in Fig. 4b calculated?

Reviewer #2 commented on the authors reply to Reviewer #3 report:

Reviewer #2 (Remarks to the Author):

I read the review report today, paying particular attention to the comments by Reviewer #3 and the author responses to those comments. It is my impression that the authors did a sufficient job of addressing the many points that Reviewer #3 raised. I suspect that that reviewer would have been content to have the paper accepted for publication.

We thank the reviewer for their detailed attention to our manuscript and apologize that they are correct, we did not fully understand the intention behind their comment. In our newly revised version we make it clear that while we give Fong et al. full credit for their discovery of the stripe domain structure it appears that in our films stripe domain structures do in fact form in films with bottom SRO electrodes, which differs from what they found. As we write in the new version of the paper, we expect this is most probably because different deposition techniques are used and the electrostatic boundary conditions are likely to be different for the two cases.

The reviewer also questioned what evidence we have that the scattering we see is from ferroelectric domains. While we feel that the totality of evidence presented in the paper made it clear that these was the case we are delighted to be able to present new evidence we obtained while the paper has been under review where we can see that the scattering from domains responds to an electric field. This is presented as new Supplementary Figure 11 in the supplementary material.

As for how the domain sizes are calculated, this is explained in the Supplementary materials, but a specific reference to that in the main text was lacking and we now provide this.

The relevant section of the paper, which spans pages 10-11 of our manuscript, has been revised to cover all of these points.

REVIEWERS' COMMENTS:

Reviewer #2 (Remarks to the Author):

The most recent revisions to the manuscript satisfy my previous concerns and I now recommend publication. I recommend that the authors include in the manuscript the point made in their reviewer response note regarding the possible difference in "resulting electrostatic boundary conditions" when they mention that films produced by sputtering and MOCVD may differ.

Jeff Eastman

Response to reviewer comments

REVIEWERS' COMMENTS:

Reviewer #2 (Remarks to the Author):

The most recent revisions to the manuscript satisfy my previous concerns and I now recommend publication. I recommend that the authors include in the manuscript the point made in their reviewer response note regarding the possible difference in "resulting electrostatic boundary conditions" when they mention that films produced by sputtering and MOCVD may differ.

Jeff Eastman

We have followed the reviewers recommendation and added this to the manuscript. The relevant sentence now reads "This difference is most likely due to the quite different growth processes used in our experiments (sputtering) and those of Fong et al. (MOCVD), resulting in different electrostatic boundary conditions for the two cases."